# Catalytic polymer self-cleavage for CO$_2$ generation before combustion empowers materials with fire safety

Wei Luo[1], Ming-Jun Chen [1] ✉, Ting Wang[1], Jin-Feng Feng[1], Zhi-Cheng Fu[1], Jin-Ni Deng[1], Yuan-Wei Yan[2], Yu-Zhong Wang [3] & Hai-Bo Zhao [3] ✉

Polymeric materials, rich in carbon, hydrogen, and oxygen elements, present substantial fire hazards to both human life and property due to their intrinsic flammability. Overcoming this challenge in the absence of any flame-retardant elements is a daunting task. Herein, we introduce an innovative strategy employing catalytic polymer auto-pyrolysis before combustion to proactively release CO$_2$, akin to possessing responsive CO$_2$ fire extinguishing mechanisms. We demonstrate that potassium salts with strong nucleophilicity (such as potassium formate/malate) can transform conventional polyurethane foam into materials with fire safety through rearrangement. This transformation results in the rapid generation of a substantial volume of CO$_2$, occurring before the onset of intense decomposition, effectively extinguishing fires. The inclusion of just 1.05 wt% potassium formate can significantly raise the limiting oxygen index of polyurethane foam to 26.5%, increase the time to ignition by 927%, and tremendously reduce smoke toxicity by 95%. The successful application of various potassium salts, combined with a comprehensive examination of the underlying mechanisms, underscores the viability of this strategy. This pioneering catalytic approach paves the way for the efficient and eco-friendly development of polymeric materials with fire safety.

Polymeric materials have established their indispensability across critical sectors of the national economy, encompassing construction, transportation, electronics, and more[1–3]. While these materials offer significant convenience in our daily lives, their high flammability presents a significant risk to human safety and property[4–6]. The World Fire Statistics Centre (WFSC) has estimated that fire-related losses account for approximately one percent of the global GDP each year. From 2012–2021, China alone reported over 1.324 million residential fire incidents, leading to 11,634 fatalities and 6738 injuries. The Hawaii wildfires of 2023 destroyed almost 3000 structures and killed at least 115 people[7–9]. Unfortunately, relying solely on firefighting strategies proves inadequate in effectively mitigating these pervasive fire-related

risks. The compelling need arises to imbue polymers with robust fire safety attributes, devoid of compromise to their inherent characteristics, thereby establishing an active defense mechanism[10,11].

In the conventional paradigm, achieving heightened fire safety performance in polymer materials hinges on the strategic integration of specific flame-retardant elements, such as halogens, phosphorus, nitrogen, boron, and silicon[12]. To illustrate, phosphorus-based flame retardants undergo decomposition into Lewis acids, which catalyze the amalgamation of charring agents, forming a polymerized, aromatic, protective carbonaceous layer that impedes the transfer of heat and mass between the flame and the matrix[13,14]. Halogenated flame retardants engender halogen-containing active radicals, thereby

[1]Green Preparation and Recycling Laboratory of Functional Polymeric Materials, College of Science, Xihua University, Chengdu, Sichuan 610039, China. [2]Zhuzhou Times New Material Technology Co., Ltd., Zhuzhou 412007, China. [3]The Collaborative Innovation Center for Eco-Friendly and Fire-Safety Polymeric Materials, National Engineering Laboratory of Eco-Friendly Polymeric Materials (Sichuan), State Key Laboratory of Polymer Materials Engineering, College of Chemistry, Sichuan University, Chengdu, Sichuan 610064, China. ✉e-mail: cmjchem@126.com; haibor7@163.com

capturing H• and HO• radicals within the chain reaction of substrate combustion and depriving the flame of sustenance[15]. However, significant challenges persist within this paradigm. These flame retardants primarily target the combustion and cracking of polymers, resulting in a generally limited flame-resistant effect. The excessive addition of these flame retardants inevitably compromises the overall material properties, particularly their mechanical properties. Moreover, the high-temperature decomposition of these retardants, which includes phosphorus and halogen-containing small molecules, can exacerbate smoke toxicity issues. For instance, when considering the application of flexible polyurethane foam (FPUF) in everyday products, achieving a high level of fire safety with a traditional approach is often challenging, even when aiming for a high oxygen index ($\geq 26\%$). The urethane bond begins to break down at only about 250 °C, and subsequently, ~90% of FPUF's mass is decomposed into flammable gases, including hydrocarbons, ethers, carbonyls etc[16,17]. Meanwhile, the combustion of polyurethane produces toxic gases, such as carbon monoxide (CO), oxides of nitrogen ($NO_x$), and hydrogen cyanide (HCN)[18,19]. Unfortunately, the traditional halogenated and phosphorus-based flame retardants almost increased the smoke toxicity of polyurethanes[20]. In some parts of the United States, the severe toxicity concerns associated with smoky emissions have even led to restrictions on using flame retardants. Therefore, innovative approaches are imperative to address the challenges posed by flame retardants[21–23].

Carbon dioxide ($CO_2$), a traditional fire extinguishing agent with a century-long history, has proven highly effective in swiftly extinguishing fires and preventing their escalation[24,25]. Interestingly, although nearly all polymers generate carbon dioxide during thermal oxidative combustion, it falls short in providing flame retardancy to materials. This limitation stems from the fact that $CO_2$ is released later in the combustion process, and the quantity released is insufficient to meet the requirements for flame extinguishment. If the polymer decomposition process can be altered to release $CO_2$ quickly and in substantial quantities at the onset of thermal decomposition rather than combustion, achieving self-extinguishing properties and reducing smoke toxicity becomes possible. Some reports have affirmed that alkali metal salts are conducive to polymer recovery in the solvent through low-temperature pyrolysis, resulting in the release of a significant amount of $CO_2$[26,27]. However, there is currently no documented instance of smart catalytic polymer self-pyrolysis enabling the early release of $CO_2$ to achieve enhanced fire safety.

Benefiting from the excellent fire-extinguishing properties of $CO_2$, we introduce a creative and highly effective flame-retardant method that employs catalytic polymer self-cleavage to release $CO_2$ proactively. Our research demonstrates that incorporating specific organic carboxylate potassium salts (K-salts) leads to a significant transformation in the pyrolytic behavior of conventional FPUF materials. This alteration facilitates the rearrangement of urethane and polyether segments within a narrower temperature range (260–320 °C) before the onset of intense decomposition, resulting in the rapid generation of a substantial volume of $CO_2$, while reducing the concentration of toxic gases (Fig. 1). When exposed to flame, FPUF can swiftly self-extinguish to achieve high fire safety through catalytic $CO_2$ generation. The inclusion of just 1.05 wt% K-formate can significantly raise the limiting oxygen index (LOI) of FPUF to 26.5%, increase its time to ignition (TTI) during large-scale fires by a remarkable 927%, and tremendously reduce the general conventional index of smoke toxicity by 95%. Moreover, it enhances the tensile strength of FPUF without compromising its toughness. This pioneering research paves the way for the environmentally conscious development of polymeric materials that offer excellent fire safety performance.

## Results
### Fire-safety performance
To illustrate the catalytic flame-retardant approach, we employed a range of K-salts paired with various anions (including sulfate, carbonate, formate, acetate, oxalate, succinate, malate and tartrate) possessing differing nucleophilic properties. These combinations were assessed to determine their respective roles and mechanisms in catalyzing the production of $CO_2$ from polyurethane. The K-salts are pre-mixed with polyether polyols prior to the in-situ foaming process of FPUF (Supplementary Fig. 1). In comparison to the highly flammable pure FPUF, which exhibits a low limiting oxygen index (LOI) value of 18% (Fig. 2a), the incorporation of K-salts (at levels below 3 wt%) leads to a significant enhancement in LOI. Notably, when filled with K-formate (1.05 wt%) and K-malate (1.31 wt%), FPUF have no melting droppings during ignition and can be immediately extinguished after removing the igniter, demonstrating impressive LOI values of 26.5% and 26.0%, respectively (Supplementary Movie 1, 2). At high oxygen concentrations of 30.5%, K-formate filled FPUF demonstrate a flame behavior that descends below the reference line along a spline, ultimately extinguishing after several intermittent flare-ups (Supplementary Movie 3). It looks as if some non-flammable gases are released in an instant to extinguish the flame. Furthermore, the horizontal burning test (UL 94-HB) reveals the high flame spread rate (63.0 ± 7.6 mm/min) of pure FPUF. While commonly used commercial expandable graphite (EG) and tris(1-chloro-2-propyl) phosphate (TCPP) at high content of 5.96 wt% slow down the flame spread rate of FPUF, they still fail to achieve the highest rating (HF-1) due to issues such as a long-burned length or melting drippings igniting the bottom cotton. In contrast, both K-formate and K-malate demonstrated the ability to self-extinguish FPUF, achieving the highest rating without any melting droppings (Supplementary Table 2 and Supplementary Movie 4–8). Most notably, when compared to a majority of previously reported flame retardants (Fig. 2b and Supplementary Tables 3, 4), both K-formate and K-malate filled FPUFs achieve significantly higher LOI values ($\geq 26\%$) and HF-1 rate in the UL 94-HB with minimal additive content (1.05 wt%), which underscores their exceptional efficiency. Interestingly, the flame retardancy of these two K-salts filled FPUFs remain consistent despite an increase in the amount (Fig. 2c). This suggests that K-salts may function as catalysts, effectively reducing the formation of flammable pyrolysis by products in FPUF.

Cone calorimetry is used to replicate real fire scenarios to further assess the fire safety of K-salt filled FPUF. An essential parameter in evaluating material fire resistance is time to ignition (TTI), which signifies the speed at which a material ignites when exposed to a heat source[28]. In Supplementary Fig. 2a and Supplementary Table 5, pure FPUF typically ignites rapidly, taking only 11 s when exposed to a flame. In stark contrast, the TTI for K-formate and K-malate filled FPUF significantly increases to 102 s and 105 s, respectively. This extension in ignition time provides invaluable additional moments for escaping and saving lives during fire incidents. In addition, we calculated the fire performance index (FPI) and fire growth index (FGI) for these materials. Generally, higher FPI values and lower FGI values indicate reduced ignitability and fire risk of materials[29]. As depicted in Fig. 2d and Supplementary Table 5, both K-formate and K-malate substantially elevate the FPI value by 600% and reduce the FGI by 40%, thereby significantly diminishing the risk of fire propagation at the early stage. This stands as the distinctive merit of the flame-retardant system.

Interestingly, the mass loss observed in FPUF filled with K-formate and K-malate prior to ignition is notably high, reaching up to 8%. This is in stark contrast to pure FPUF, which experiences only a modest 0.2% mass loss, as evident in Supplementary Fig. 2b–d. Notably, during ignition, both K-formate and K-malate filled FPUF release > 0.35 kg/kg of $CO_2$, whereas pure FPUF exhibits no $CO_2$ emission at this stage (Fig. 2e). To precisely determine the concentrations of $CO_2$ and combustible gases such as $C_2H_6$, $C_3H_8$, $CH_2O$ and $C_3H_4O$, FTIR coupled with a cone calorimeter were employed[30]. Figure 2f, Supplementary Fig. 3 and Supplementary Table 6 clearly illustrate that during combustion, $CO_2$ production significantly surpasses that of other gases. The presence of K-salts not only amplifies the $CO_2$ concentration by 72% but

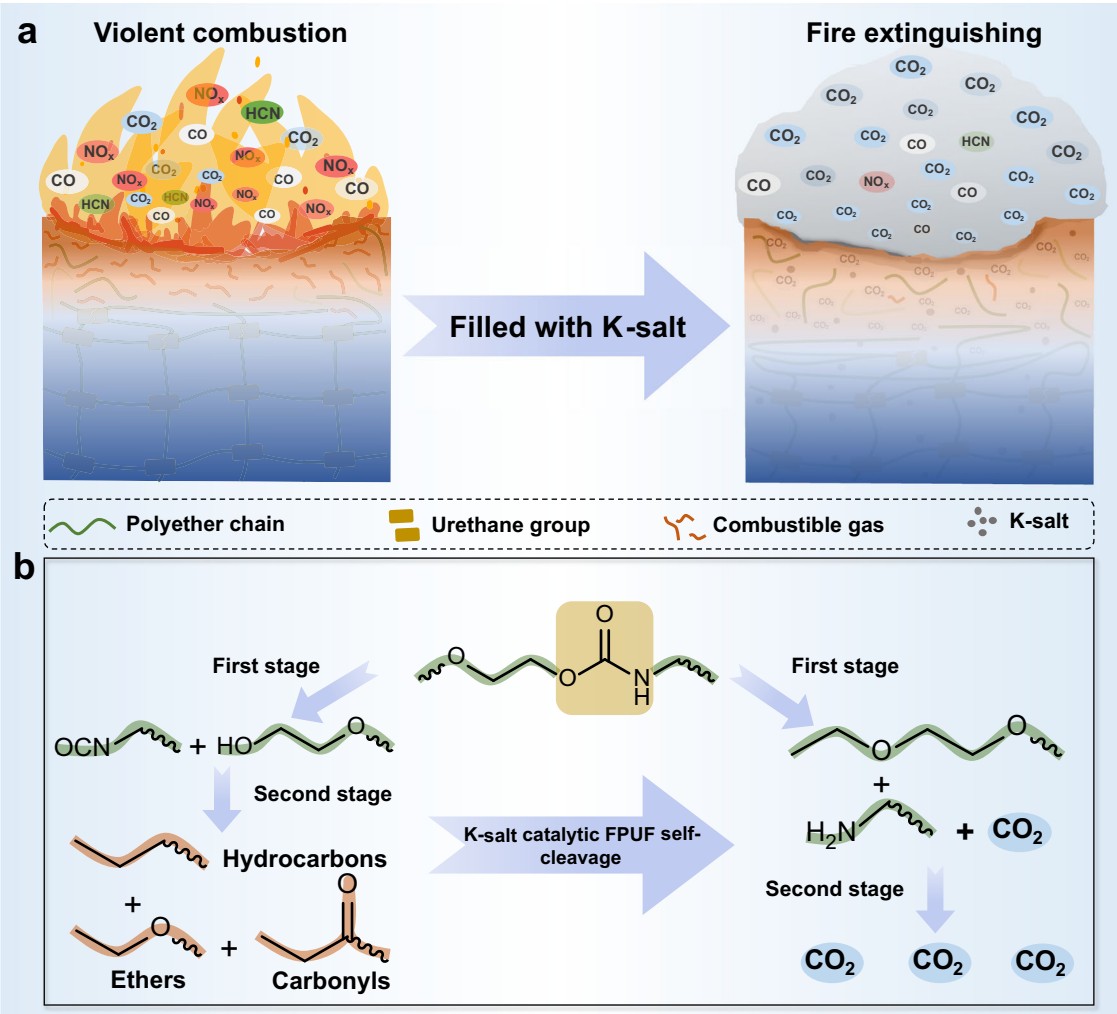

**Fig. 1 | Schematic illustration of the catalytic polyurethane self-cleavage for CO₂ generation empowering the materials with fire safety. a** Diagram illustration of K-salt catalyzed polyurethane auto-pyrolysis to release $CO_2$ early for fire extinguishing. **b** Pyrolysis processes of polyurethane catalyzed by K-salt.

also notably diminishes the levels of combustible molecules. This suggests that K-salts catalyze the thermal oxidation decomposition of polyurethane, resulting in substantial $CO_2$ generation both before and after ignition. In essence, K-salts appear to act as catalysts, expediting the production of a significant quantity of $CO_2$ from polyurethane before combustion, empowering materials with fire safety to effectively mitigate the risk of fire propagation.

More importantly, this catalytic polymer self-cleavage strategy can greatly suppress the smoke and toxic gases that cause 50–80% of deaths in fire disasters[31]. Polyurethane combustion produces harmful gases like CO, $NO_x$, and HCN. Figure 2g illustrates that TCPP addition notably elevates CO and HCN production. Conversely, both K-formate and K-malate substantially reduce CO, HCN, and $NO_x$ concentrations. Particularly, K-salt filled FPUFs exhibit over 60% reduction in HCN production. K-formate leads to a remarkable decrease in $NO_x$ concentrations from 96.3 ppm – 1.8 ppm, well below the immediately dangerous to life and healthy (IDLH) concentration of $NO_x$ (20 ppm) per the National Institute of Occupational Safety and Health (NOISH)[32]. The conventional toxicity index (CIT$_G$) of FPUFs filled with K-formate and K-malate decreases by 95% and 86%, respectively, demonstrating the excellent smoke toxicity inhibitory effect of K-salts (Fig. 2h and Supplementary Table 7). Additionally, as indicated in Supplementary Table 8, the incorporation of K-salts leads to a substantial reduction in smoke density (~20) in FPUF. This is notably lower compared to pure foam (83) or foam with TCPP (107). With a high LOI value, low flame

spread rate, and diminished smoke toxicity and density, K-salts filled FPUF exhibits outstanding fire safety.

## Pyrolysis behavior

The pyrolysis behaviors of different K-salts filled FPUFs are evaluated by thermogravimetric analysis (TGA). Under an air atmosphere, the TGA curves for K-salt filled FPUF differ significantly from those of pure polyurethane, as shown in Fig. 3a–e and Supplementary Table 9. Most K-salt filled samples exhibit a sharp mass loss within a narrow temperature range of 260–320 °C, except for K-sulfate and K-oxalate. This clearly indicates an interaction between K-salts and FPUF during thermal oxidation decomposition. It can be observed (Fig. 3a and Supplementary Table 9) that the primary decomposition process of pure FPUF can be divided into two stages. The first stage involves the breakage of the hard urethane segment at 260–320 °C, resulting in a mass loss of ~26%. The second stage entails the decomposition of the soft polyether segment at 340–450 °C, with a mass loss of around 41%. The effect of different K-salts on the thermal decomposition process of polyurethane varies, with K-sulfate and K-carbonate filled FPUF being the first to be evaluated. In comparison to K-sulfate, K-carbonate leads to a reduction of the initial decomposition temperature ($T_{5\%}$) and the maximum weight loss temperature ($T_{max}$) by 26 °C and 102 °C (Fig. 3a and d, e), respectively. The mass loss of K-carbonate filled FPUF during 260–320 °C is as high as 76%, significantly exceeding those of pure FPUF (26%) and K-sulfate filled FPUF (19%). Different anions paired with

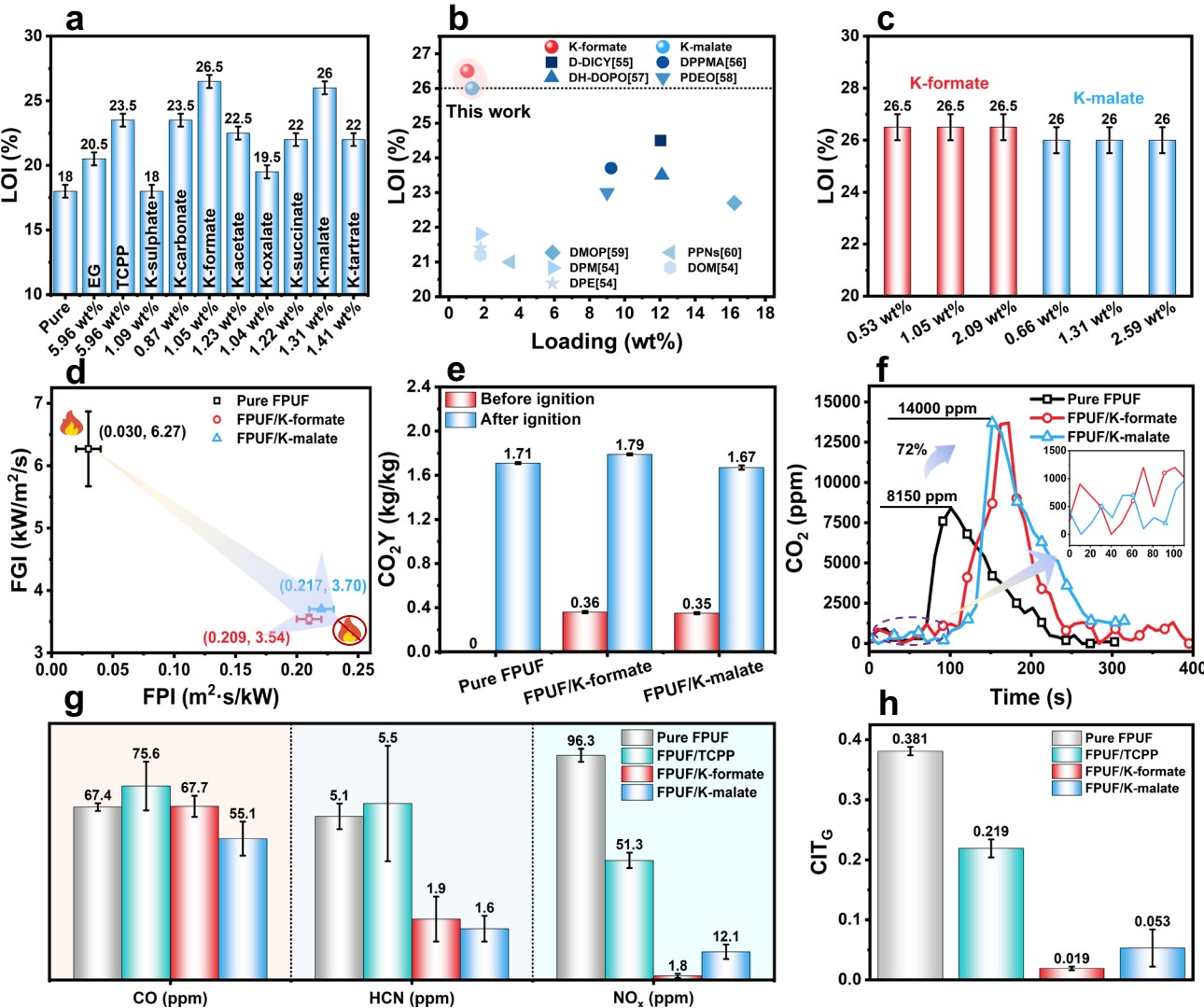

**Fig. 2 | Fire-safety performance. a** Bar graphs of LOI values for expandable graphite (EG), tris(1-chloro-2-propyl) phosphate (TCPP), and K-salts filled FPUF. **b** Comparison of LOI values from this study with the recent works[54–60]. **c** Bar graph of LOI values for FPUF with varying additions of K-formate and K-malate. **d** Fire performance index (FPI) and fire growth index (FGI) of K-formate and K-malate filled FPUF. **e** Bar graphs of carbon dioxide yield ($CO_2Y$) before and after ignition of K-formate and K-malate filled FPUF under cone calorimeter test. **f** Concentrations of $CO_2$ obtained from cone calorimeter coupled with FTIR. **g** Concentrations of $CO_2$, HCN and $NO_x$ in the smoke toxicity test. **h** General conventional index of toxicity ($CIT_G$) obtained from smoke toxicity tests. All error bars represent a particular confidence interval.

potassium exhibit distinct activities in promoting the breakage of urethane and polyether segments in polyurethane. The anions in alkali metal salts act as nucleophiles, attacking urethane groups and catalyzing the depolymerization of polyurethanes[33,34]. Nucleophilicity mainly depends on the alkalinity and steric hindrance of additives[35,36]. Since the Lewis alkalinity of carbonate is higher than that of sulfate, K-carbonate exhibits better nucleophilicity, resulting in the more violent decomposition of polyurethane.

The Fig. 3b and d, e clearly demonstrates that organic K-salts such as K-formate, K-acetate, and K-succinate notably accelerate the decomposition of FPUF within the 260–320 °C range. However, K-oxalate stands as an exception, as the decomposition curve for K-oxalate filled FPUF closely resembles that of pure FPUF. This observation underscores the significant influence of the carboxylate structure of potassium salts on the catalytic activity of polyurethane decomposition. Specifically, K-formate exhibits stronger nucleophilicity than K-acetate due to its smaller steric hindrance. As steric hindrance continues to increase, the nucleophilicity of K-oxalate and K-succinate weakens. Notably, K-oxalate exhibits the lowest catalytic

activity, as the electron absorption of the ortho-carbonyl group further diminishes nucleophilicity. Furthermore, we compared K-malate and K-tartrate with K-succinate to evaluate the effect of hydroxyl groups. It has been reported that K+ (acting as a Lewis acid) tends to combine with RO- (serving as a Lewis base) to form potassium alcoholates, which exhibit stronger nucleophilicity than potassium carboxylates[37,38]. This may explain why, despite having larger steric hindrance than K-succinate, both K-malate and K-tartrate promote rapid mass loss in FPUF within the 260–320 °C range, as shown in Fig. 3c–e. These results indicate that K-salts with stronger nucleophilicity display superior catalytic activity in the decomposition of polyurethane, especially within the 260–320 °C range. This is a key factor contributing to the improvement of FPUF flame retardancy (Fig. 3f).

**Mechanism of catalytic decomposition for $CO_2$ release**

We further unveil the chemical mechanism of K-salts on catalyzing the early decomposition of polyurethane through TGA coupled with FTIR (TGA-FTIR), TGA coupled with MS (TGA-MS), and TGA coupled with DSC (TGA-DSC). The pyrolysis products of K-salts filled FPUF

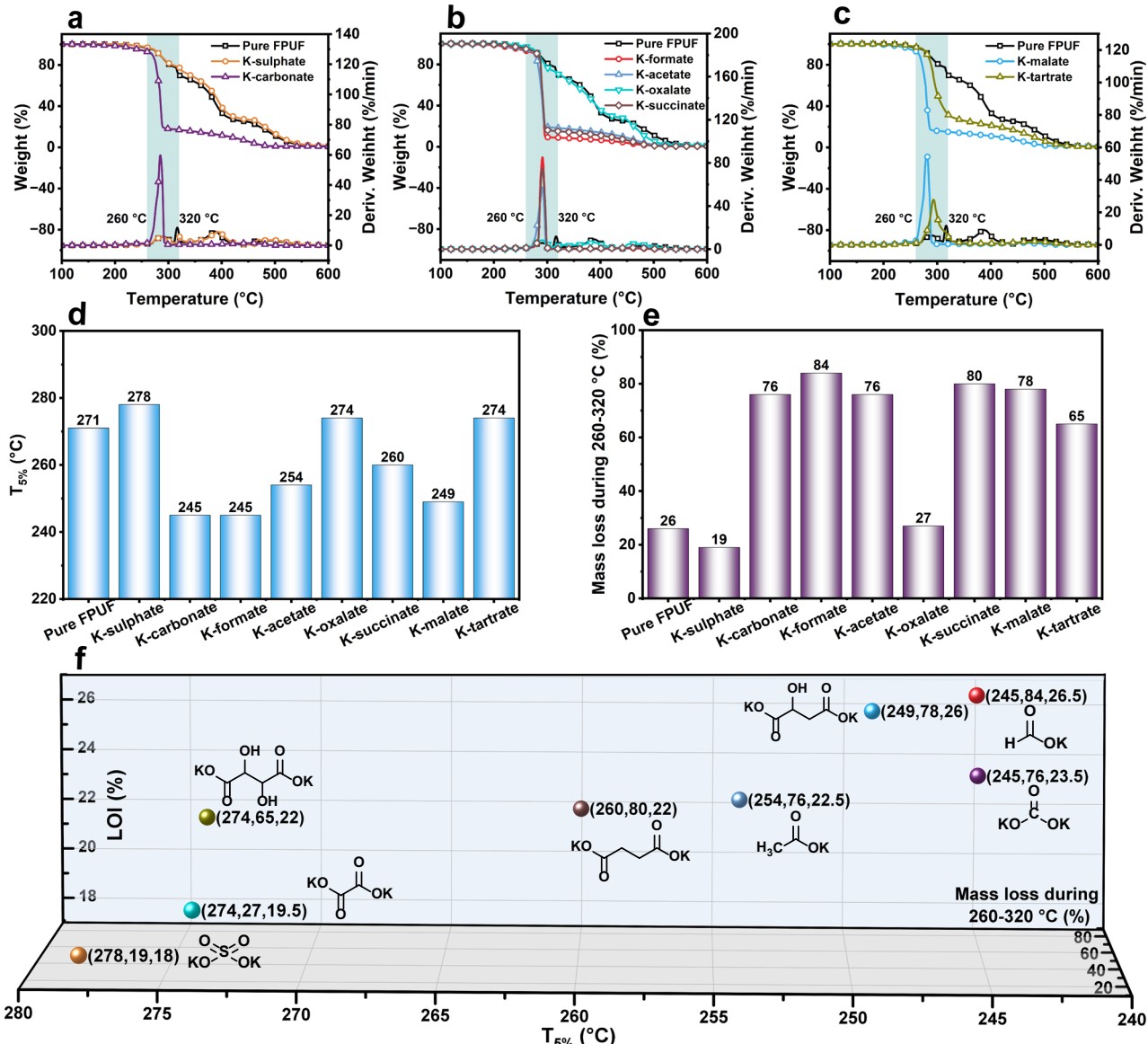

**Fig. 3 | Pyrolysis behavior under air atmosphere. a–c** TGA and DTG curves of K-salts filled FPUF. **d** The bar graphs of initial pyrolysis temperature ($T_{5\%}$) of K-salts filled FPUF. **e** The bar graphs of the mass loss of K-salts filled FPUF during 260–320 °C. **f** Three-dimensional distribution of $T_{5\%}$, mass loss during 260–320 °C, and LOI values of K-salts filled FPUF (x, y, z = $T_{5\%}$, mass loss during 260–320 °C, LOI, respectively).

are firstly analyzed by TGA-FTIR under air atmosphere. It reveals that potassium salts, characterized by their notable nucleophilic properties, accelerate the cleavage of urethane groups at ~200 °C. This process is accompanied by the release of carbon dioxide ($CO_2$), as evidenced by distinct peaks in the FTIR spectra at 2300–2400 $cm^{-1}$. The presence of potassium salts has a negligible impact on the diversity of pyrolysis products in comparison to those observed in the polyurethane precursor polyether, which constitutes 65% of the polyurethane composition[39]. However, it is essential to highlight that K-carbonate, K-formate, K-acetate, K-succinate, K-malate, and K-tartrate exert a significant influence by markedly elevating the relative release of $CO_2$ (Fig. 4a–c and Supplementary Fig. 4). Among these, K-formate and K-malate filled FPUF exhibit the highest $CO_2$ absorbance levels within the temperature range of 260–320 °C (Supplementary Fig. 5). A quantitative analysis presented in Supplementary Fig. 6 and Supplementary Table 10 involves the determination of cumulative intensity by integrating the absorbance curve over the temperature interval from 260 °C – 320 °C[40]. Remarkably, the $CO_2$ cumulative intensity for FPUFs impregnated

with K-formate and K-malate soared to 1912 a.u./g and 1762 a.u./g, respectively (Fig. 4d). These values markedly surpassed those recorded for other foam samples, underscoring the exceptional catalytic impact of these potassium salts.

TGA-MS and TG-DSC results reveal the chemical processes through which K-formate and K-malate catalyze the conversion of polyether chains into $CO_2$. TGA-MS shows that at around 300 °C, a significant amount of $CO_2$ (m/z 44) is released, concurrently with substantial amounts of $H_2$ (m/z 2) and $H_2O$ (m/z 18). Research suggests that a hydrogen-rich environment is conducive to the complete decomposition of polyether chains (Supplementary Figs. 7, 8). Simultaneously, significant quantities of formaldehyde (m/z 28 CO and m/z 29 HCO•) and acetaldehyde (m/z 43 $CH_3CO•$) fragments are detected after hydrogen supply, indicating that intermediates such as formaldehyde, containing active hydrogen, play a vital role in the conversion of polyether chains into $CO_2$ (Supplementary Fig. 9)[41–43]. In TGA-DSC analysis, a distinct exothermic peak is evident around 300 °C for both K-formate and K-malate filled FPUF, whereas pure FPUF exhibits a milder exothermic peak at 350 °C (Fig. 4e). This observation

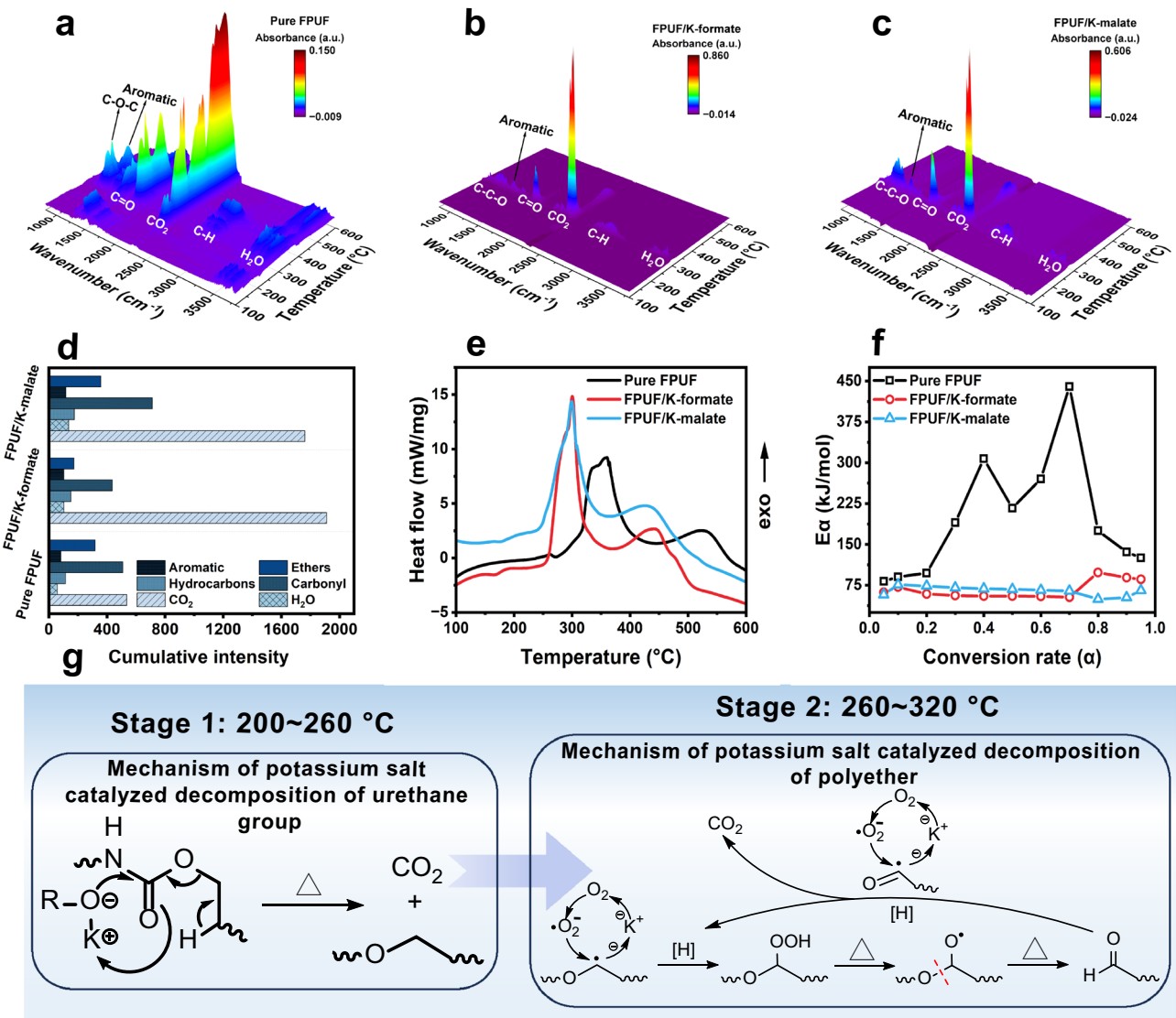

**Fig. 4 | Mechanism of catalytic decomposition for CO₂ release. a–c** The three-dimensional spectra of pyrolysis products of K-formate and K-malate filled FPUF collected from TGA coupled with FTIR under air atmosphere. **d** Gaseous phase products cumulative intensity under air atmosphere at 260–320 °C, and the cumulative intensity was obtained by integrating the normalized absorbance curve of K-salt filled FPUF during 260–320 °C. **e** DSC curves of K-formate and K-malate filled FPUF under air atmosphere. **f** Thermal decomposition average apparent activation energy (Eα) calculated from Kissinger-Akahira-Sunose (KAS) and Flynn-Wall-Ozawa (FWO) methods. **g** Mechanism of CO₂ generation from early decomposition of polyurethane catalyzed by K-salt.

suggests that the polyurethane chains undergo an accelerated thermal oxidation reaction when catalyzed. To deduce the catalytic mechanism behind the thermal oxidation reaction, two widely used non-model methods, namely, Kissinger-Akahira-Sunose (KAS) and Flynn-Wall-Ozawa (FWO), are conducted to calculate the apparent activation energy (Eα) of K-formate and K-malate filled FPUF (Supplementary Figs. 10, 11)[44,45]. The coefficients of determination (R²) for the Arrhenius curves fitted using the KAS and FWO models are mostly >0.9 (Supplementary Table 11). And the apparent activation energies (Eα) calculated by both the KAS and FWO methods exhibit close agreement (Supplementary Fig. 12), indicating the consistency between these two models. As depicted in Fig. 4f, the average Eα for pure FPUF within a conversion rate range of 0.2 – 0.8 is -242 kJ/mol, representing the energy required for the thermal oxidative decomposition of polyether segments[39]. In contrast, the average Eα for K-formate and K-malate filled FPUF is significantly reduced to 61 kJ/mol and 65 kJ/mol, respectively. This unequivocally demonstrates the high catalytic activity of K-salts in the thermal oxidation decomposition of polyether segments.

Based on the aforementioned findings, the mechanism by which K-salts catalyze the early decomposition of polyurethane to produce CO₂ can be elucidated through two stages (Fig. 4g). In the initial stage (200–260 °C), K-salts, particularly those with strong nucleophilicity, facilitate the cleavage of the urethane group at lower temperatures, around 200 °C. RCOO- initiates an attack on the electron-deficient carbonyl carbon, while K+ draws the carbonyl electron cloud to form a six-membered ring intermediate[33,46,47]. The shifting of the electron cloud within the carbonyl group promotes the decomposition of the urethane group, leading to the formation of carbamic acid. This carbamic acid is readily decomposed into CO₂ and primary amines upon heating (Supplementary Fig. 13a)[16]. In the case of K-malate containing hydroxyl group, potassium alcoholate with stronger nucleophilicity typically precedes its attack on the electron-deficient carbonyl carbon. This further enhances the promotion of urethane group pyrolysis into carbamic acid, ultimately generating CO₂ (Supplementary Fig. 13b)[37,38,48]. Although the release of CO₂ in the first stage is relatively low, the early decomposition of urethane groups results in the formation of polyurethane

precursors, particularly polyether, in a molten state. This state is crucial for the subsequent thermal oxidation of polyether catalyzed by potassium salts.

During the second stage (260–320 °C), the catalyzed decomposition of polyether by potassium salts unfolds in three distinct steps[49], as illustrated in Supplementary Fig. 14. Firstly, polyether undergoes pyrolysis to form alkyl radicals (R•). Some of these alkyl radicals have their electrons attracted by $K^+$, leading to the creation of reactive oxygen species when they combine with oxygen from the gas phase[50]. Subsequently, these reactive oxygen species join forces with alkyl radicals to generate peroxy radicals (ROO•), capable of capturing active hydrogen to produce unstable hydroperoxide (ROOH). ROOH plays a pivotal role in the thermal oxidative decomposition of polyether chains, as its cleavage yields polyether alkoxy radicals (RO•) and HO•. The breaking of RO• proceeds along two primary pathways: one involves the capture of the hydrogen on the ortho carbon of RO• by HO•, yielding water and ester; the other entails the rupture of the C-O bond in RO•, leading to the formation of new RO• and aldehyde. Ultimately, esters readily decompose into olefins and carboxylic acids, serving as a source of $CO_2$. Aldehydes are effective hydrogen donors for hydroperoxide formation, which is also a source of $CO_2$[49]. In addition, once aldehydes have exhausted their active hydrogen, $K^+$ catalyzes the production of peroxy acids. These peroxy acids then engage in Bayer-Villiger oxidation reactions with aldehydes to generate carboxylic acids, which are subsequently decarboxylated to produce $CO_2$[51]. Consequently, both K-formate and K-malate effectively and quickly catalyze FPUF to spontaneously generate a large amount of $CO_2$ in the early stage of heating (260–320 °C). This successfully empowers the material with fire safety, effectively preventing materials from igniting.

## Mechanical and comprehensive performance

In contrast to conventional flame retardants, which often compromise the mechanical properties of materials, the catalytic flame-retardant approach offers a distinct advantage by endowing FPUF with excellent overall performance characteristics. Scanning electron microscopy (SEM) images (Supplementary Fig. 15) reveal that K-formate and K-malate exhibit good compatibility and dispersibility within the FPUF matrix. This results in their filled FPUFs retaining a continuous open-cell structure and a smooth cell skeleton surface. The preservation of the cellular structure is crucial for maintaining the mechanical performance of FPUF. Furthermore, the catalytic polymerization of polyurethane chains and the formation of ion clusters by K-salts contribute to the enhancement of the strength and elasticity of polyurethane[52,53]. As demonstrated in Fig. 5a and Supplementary Fig. 16, and Supplementary Table 12, the tensile strength of K-formate and K-malate filled FPUF is improved by 42% and 19%, respectively, compared to that of pure FPUF. Interestingly, K-formate and K-malate not only increase fracture strength but also exhibit remarkable improvements in tensile toughness. The corresponding tensile toughness reaches 52 kJ/m$^2$ and 35 kJ/m$^2$, representing improvements of 108% and 40%, respectively (Fig. 5b). Figure 5d–f shows that K-salts filled FPUF exhibit rougher fracture surfaces, characterized by larger deformations and more complex fracture paths. This indicates that K-salts effectively reduce the transfer of mechanical stress in FPUF under high strain conditions[54]. It is noted that the negative impact of K-salts on the elastic resilience of FPUF is negligible. This is evident from the fact that the compressive stress and deformation recovery of K-formate and K-malate filled FPUF can be maintained at over 90% even after 100 compression cycles (Fig. 5c, Supplementary Fig. 17 and Supplementary Table 12).

To emphasize the merits of our proposed catalytic polymer auto-pyrolysis approach for $CO_2$ release as a flame-retardant strategy, we compared FPUF/K-formate with other recently reported flame-retardant FPUFs with comprehensive performance. As illustrated in Fig. 5g and detailed in Supplementary Tables 3, 4, our strategy stands out due to its remarkable flame retardancy achieved at the lowest loading, surpassing the highest reported flame-retardant efficiency to date. K-formate not only significantly enhances the tensile stress of FPUF but also exhibits the most substantial improvement in terms of tensile strain. Perhaps most importantly, in the context of today's increasingly pressing environmental concerns, our strategy offers a significant boost to the fire safety performance of FPUF without introducing any harmful flame-retardant elements, such as phosphorus and halogen. This represents a notable achievement in enhancing the fire safety performance of polymer materials while adhering to stringent environmental considerations. In summary, our strategy excels in flame retardant efficiency, enhances material mechanical properties, and promotes environmental protection, underscoring its intrinsic advantages.

## Discussion

In summary, we have introduced a pioneering catalytic flame-retardant strategy that utilizes catalytic polymer auto-pyrolysis to actively release $CO_2$. This unique green approach enables polymers to self-extinguish fires at an exceptionally early stage, akin to empowering materials with fire safety, all without the need for traditional flame-retardant additives. Our research has uncovered that the use of K-salts, known for their strong nucleophilic properties, induces a more efficient rearrangement of the urethane and polyether segments within traditional FPUF materials. This rearrangement occurs within a narrower temperature range (260–320 °C), leading to a substantial release of $CO_2$ before any violent decomposition occurs. As a result, we can significantly enhance the flame retardancy, toxic smoke suppression, and mechanical properties of FPUF materials with an ultra-low loading of catalyst, thanks to the inherent advantages of this strategy. This method holds immense promise for widespread application, as the K-salts employed are cost-effective and readily available, and the process aligns with stringent environmental protection requirements. Our findings offer valuable insights into the development of environmentally friendly polymer materials boasting exceptional fire safety performance. Given the prevalence of carbon, hydrogen, and oxygen elements in most polymers, we anticipate that similar catalysts could yield significant improvements in fire safety across a range of polymeric materials.

## Methods

### Materials

Triethanolamine (TEOA, AR), potassium sulfate (K-sulfate), potassium carbonate (K-carbonate, AR), potassium formate (K-formate, AR), potassium acetate (K-acetate, AR), potassium oxalate (K-oxalate, GR), and potassium tartrate (K-tartrate, AR) were supplied by Kelong Chemical Reagent Corp (Chengdu, China). Potassium succinate (K-succinate, 95%) and potassium malate (K-malate, 95%) were supplied by Jiangsu Aikang Biomedical R&D Co., Ltd (Nanjing, China). Tris(1-chloro-2-propyl) phosphate (TCPP) was supplied by Zhang Jia Gang YaRui Chemical Co., Ltd. (Zhangjiagang, China). Expanded graphite (EG) was supplied by Qingdao Tianhe Graphite Co., Ltd. (Qingdao, China). Polyether polyols (EP-330N, number average molecular weight of 4800, average functionality of 3.0, -OH content of 35 mg of KOH/g) were obtained from Zibo Dexin Federal Chemical Industry Co., Ltd. (Zibo, China). Modified 4,4' diphenylmethane diisocyanate (MDI-2412, -NCO content of 26.5%) was obtained from Huntsman Chemical Trading Co., Ltd. (Shanghai, China). The catalysts (70% bis(dimethylaminoethyl) ether, A-1; 33% triethylenediamine solution, A33B) and surfactant (DC-2525) were kindly supplied by Tianjin Qiyue New Materials Co., Ltd. (Tianjin, China). Deionized water was used as a chemical blowing agent. All reagents were used as received.

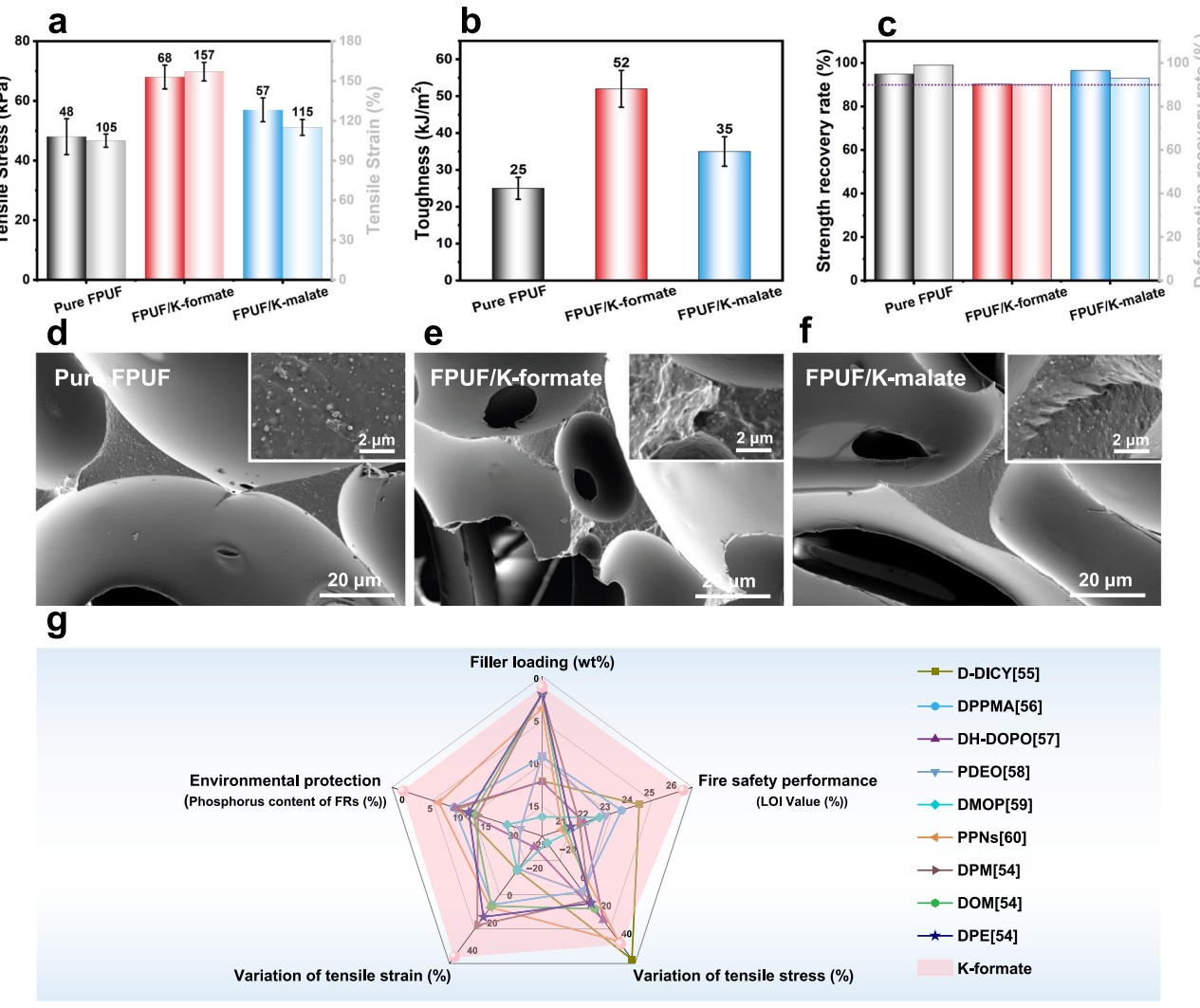

**Fig. 5 | Mechanical and comprehensive performance. a** Tensile stress, strain, and **b** toughness. **c** Compression resilience. **d**–**f** SEM images for the tensile fracture of pure FPUF, PUFF/K-formate, FPUF/K-malate, respectively. **g** Comparison of K-formate filled FPUF with other recently reported flame-retardant FPUF samples in terms of filler loading, fire safety performance, tensile stress variation, tensile strain variation, and environmental protection[54–60]. All error bars represent a particular confidence interval.

## Preparation of potassium salt filled FPUF

FPUF was prepared by a one-pot and free-rise foaming method, and the NCO index (molar ratio of NCO/OH) of all foams is controlled at 0.85 (Supplementary Fig. 1). The formulations are shown in Supplementary Table 1. Potassium salt (K-salt), polyether polyol (EP-330 N), deionized water, triethanolamine (TEOA), catalysts (A-1, A33B) and surfactant (DC-2525) were added to a beaker and stirred vigorously for 2 min. Subsequently, MDI-2412 was added with vigorous stirring for 5–8 s, and the mixture was immediately poured into a cast mold. The samples were further cured for 72 h under ambient conditions. The density of all FPUF samples was controlled at 60 ± 2 kg/cm³.

## Characterizations

The limiting oxygen index (LOI) was measured by a JF-3 oxygen index meter (Nanjing Jiangning Analytical Instruments Co., Ltd., China) according to ASTMD-2863, and the dimensions of the specimen were 10 mm × 10 mm × 150 mm. The average values of five samples were recorded. The horizontal burning test (UL 94-HB) was performed on a CZF-3 instrument (Nanjing Jiangning Analytical Instruments Co., Ltd., China) according to ISO 9772–2020, and the dimensions of the specimen were 150 mm × 50 mm × 13 mm. The average values of five samples were recorded. A cone calorimeter device (Fire Testing

Technology, UK) was used to evaluate the combustion performance of the FPUF filled with potassium salt according to ISO 5660-1. The specimens were exposed to a heat flux of 25 kW/m² with a size of 100 mm × 100 mm × 25 mm. Each group of samples was tested three times in parallel, and the results were averaged. The smoke density test was measured by a FTT0064 smoke density box (Fire Testing Technology, UK) according to ISO 5659-2. The specimens were exposed to a heat flux of 25 kW/m² with a size of 75 mm × 75 mm × 25 mm. Each group of samples was tested three times in parallel, and the results were averaged. The smoke toxicity gas concentration test was measured by a FTT0095 smoke toxicity tester (Fire Testing Technology, UK) according to ISO 5659-2. Each group of samples was tested three times in parallel, and the results were averaged. The thermal stability of the samples was tested by TGA-550 thermogravimetry (TA Instruments, USA). The samples (5–8 mg) were heated from 40 °C to 650 °C at different heating rate (10 °C/min, 15 °C/min, 20 °C/min, 25 °C/min) under air atmosphere with gas flow of 50 mL/min. The pyrolysis products were monitored by TGA coupled with FTIR (TGA-FTIR), TGA coupled with MS (TGA-MS), and TGA coupled with DSC (TGA-DSC), respectively. As for TGA-FTIR testing, a TGA-550 (TA, USA) was combined with a Nicolet IS10 (Thermo-Fisher Scientific, USA) infrared spectrometer. The samples (1–3 mg) were heated from 40 °C to 650 °C

at the heating rate of 10 °C/min under air atmosphere with gas flow of 100 mL/min. For TGA-MS testing, the gaseous products were analyzed by Thermo Plus EV2/Thermo Mass Photo (Rigaku, Japan) under the same conditions as TGA-FTIR. The simultaneous TGA-DSC apparatus of the Germany Netzsch STA 449 F3 type was used to test the endothermic and exothermic behavior of the samples. The testing conditions were similar to TGA-FTIR and TGA-MS, except for the gas flow rate that was 50 mL/min for TGA-DSC testing. To eliminate the error caused by the mass difference of the tested samples during the TGA-FTIA and TGA-MS test process, in this work, we normalized the measured absorbance by mass. Therefore, the normalized absorbance of each sample is calculated by dividing the measured absorbance by its own mass. The density was calculated by the volume and weight of potassium salt filled FPUF according to ISO 845: 2006. The size of each specimen was no less than 100 cm³ and the average values of five samples were recorded. The mechanical properties of potassium salt filled FPUF were determined by the tensile measurements according to ISO 1798:2008. The tensile speed was 500 mm/min with 250 N load cells. The compression and resilience properties were characterized by a universal testing machine (Instron 5400, USA) equipped with 500 N load cells at a 50% compression deformation and a strain rate of 20 mm/min. The microstructures of the brittle fracture and tensile fracture were observed by scanning electron microscopy (SEM, ZEISS Sigma 300) at a 3 kV acceleration voltage. Samples were over gilded using an Eiko IB-3 ion coater instrument before test.

### Kinetic model

The fundamental rate equation used in all kinetics studies is generally described as:

$$\frac{d\alpha}{dt} = kf(\alpha) \tag{1}$$

where k is the rate constant and f(α) is the reaction mode. Equation (1) expresses the rate of conversion, dα/dt, at a constant temperature as a function of the reactant conversion loss and rate constant. In this study, the conversion rate α is defined as:

$$\alpha = \left(\frac{w0 - wt}{w0 - wf}\right) \tag{2}$$

where $w_t$, $w_0$ and $w_f$ are the time t, the initial and final weights of the sample, respectively. The rate constant k is generally given by the Arrhenius equation:

$$k = A \exp\left(\frac{-E\alpha}{RT}\right) \tag{3}$$

where Eα is the apparent activation energy (kJ/mol), R is the gas constant (8.314 J/K mol), A is the pre-exponential factor (min⁻¹), and T is the absolute temperature (K). The combination of Eqs. (1) and (3) gives the following relationship:

$$\frac{d\alpha}{dt} = A \exp\left(\frac{-E\alpha}{RT}\right)f(\alpha) \tag{4}$$

For a dynamic TGA process, including the heating rate, $\beta$ = dT/dt, into Eq. (4), Eq. (5) is obtained as:

$$\frac{d\alpha}{dt} = \left(\frac{A}{\beta}\right)\exp\left(\frac{-E\alpha}{RT}\right)f(\alpha) \tag{5}$$

Equations (4) and (5) are the fundamental expressions of analytical methods to calculate kinetic parameters on the basis of TG data.

The Kissinger-Akahira-Sunose (KAS) and Flynn-Wall-Ozawa (FWO) methods were used in this research, as shown in Eqs. (6) and (7). As a representative model of the model-free kinetic model, it had high applicability to polymer pyrolysis.

$$\ln\left(\frac{\beta}{T^2}\right) = \ln\left[\frac{AR}{g(\alpha)E}\right] - \frac{E}{RT} \tag{6}$$

where the g (α) of the KAS method is the integral mechanism of the degradation reaction. When α is certain, g (α) is also certain. The KAS method makes a linear relationship curve with ln(β/T²) to 1/T and then performs linear fitting. The corresponding kinetic parameter E can be obtained from the slope of the straight line.

$$\log\beta = \log\left[\frac{AE}{Rg(\alpha)}\right] - 2.315 - 0.4567\frac{E}{RT} \tag{7}$$

where the FWO method makes a linear relationship curve with log(β) to 1/T and then performs linear fitting. The corresponding kinetic parameter E can be obtained from the slope of the straight line.

## Data availability

All data supporting the findings of this study are available within the article, as well as the Supplementary Information file, or available from the corresponding authors upon request.

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

## Acknowledgements

Financial support by the National Natural Science Foundation of China (21975208, U22A20150 and 22205184) and the Sichuan Science and Technology Program (2023ZYD0030 and 2020JDJQ0062) are sincerely acknowledged.

## Author contributions

W. Luo: conceptualization, sample preparation, data analysis, investigation, writing-original draft preparation. M.-J. Chen: conceptualization, methodology, supervision, formal analysis, writing-review and editing; T. Wang: methodology, formal analysis. J.-F. Feng. investigation, data analysis. Z.-C. Fu and J.-N. Deng: data analysis. Y.-W. Yan: smoke density and toxicity test and analysis; Y.-Z. Wang: supervision, methodology; H.-B. Zhao: conceptualization, formal analysis, writing-review and editing.

## Competing interests

The authors declare no competing interests.
