## [Peer Review File · Nature Communications]

Catalytic Polymer Self-Cleavage for CO₂ Generation Before Combustion Empowers Materials with Intelligent Fire SafetyReviewers' Comments:

Reviewer #1:

Remarks to the Author:

This paper describes an approach to improving the flame retardancy of polyurethane foams by incorporating compounds that accelerate the decomposition of the urethane bond as well as oxidation of polyether polyols to generate carbon dioxide that extinguishes fire.

Overall, the approach is very creative and doesn't rely on halogenated or phosphorous compounds. The authors have done significant experimental work to verify the reaction mechanisms and the conclusions are supported by the experimental data.

A few issues to consider prior to publication:

The "fire extinguisher" analogy seems weak and not especially scientific. A CO₂ fire extinguisher has pressurized CO₂ that is released. This system chemically generates CO₂. So other than CO₂ being emitted the analogy isn't very strong. I'd suggest removing it from the title and Figure 1.

There are a number of phrases in English that are a bit awkward.

In the abstract (lines 32-33): "The incorporation of just 1.68 parts per hundred polyether polyols by weight..." This took me a while to figure out. Suggest: "The incorporation of just 1.68 parts per hundred by weight relative to polyether polyol of potassium formate..."

Line 55: should be "Lewis acids" (since retardants is plural, the acids has to be plural)

Lines 62-63: Suggest: "The excessive addition of these retardants may compromise the overall material properties, particularly their mechanical properties."

Line 91: "achieve high fire safe" Not sure what this means.

A major issue that should be remedied is that I was very confused at the outset as to what was actually being done. Figure 1a is very far from the section in the introduction that gives an overview. And then the results section just jumps in discussing the experiments. I think the authors should have a standalone figure (not part of figure 1) that gives a concrete overall perspective on what is being done to give the reader the proper context so they can understand what experiments are being carried out.

Line 104: "The K-salts are introduced into the FPUF matrix during the in-situ foaming process." Not sure I understand how this is done. I would think the additives need to be mixed into the formulation prior to the foaming process.

Line 123: Please define TTI.

Reviewer #2:

Remarks to the Author:

The authors present an interesting way to improve fire safety of PU foams by simple incorporation of K salts. The manuscript benefit from the following suggestion:

1) As LOI is not a standard test for PU foam applications, I suggest to test the foams using UL94 HB

test which is most common for automotive applications. Furthermore, LOI tests are very misleading as melting away of the foam from the applied flame can give wrong LOI values.

2) I am not sure if the authors can really claim that the fire performance is better for the modified foams, just from LOI values. In fact, the cone experiments provided in the supplementary info paint a different picture. The modified foams have higher HRR which indicates a higher risk in case of fire.

3) please provide information such as smoke density and smoke toxicity (considering ammonia, HCN, and carbon monoxide evolution) for the PU foams.

Reviewer #3:

Remarks to the Author:

Authors investigated the role of potassium salts as flame retardants for polyurethane foams acting through catalytic polymer autopyrolysis mode. Although it is claimed that "Here, we introduce an innovative strategy for the first time..." the autocatalytic reactive flame retardants (ACRFR) concept has been applied for polyurethanes, e.g. *Materials Chemistry and Physics*, Volume 267, 15 July 2021, 124636; *Polymer Degradation and Stability*, Volume 177, July 2020, 109168, and references cited therein. Potassium salts, such as potassium carbonate derivatives have also been tested as PU foams flame retardants, e.g. INVESTIGATION OF EFFECTS OF ALUM AND POTASSIUM SESQUICARBONATE ON THE FIRE CHARACTERISTICS OF FLEXIBLE POLYURETHANE FOAM (<https://www.projectreserve.com/2020/04/investigation-of-effects-of-alum-and-potassium-sesquicarbonate-on-the-fire-characteristics-of-flexible-polyurethane-foam.html>). This work is methodologically sound, however the degree of novelty does not warrant publication in *Nature Communication*.

Responses to Reviewer's comments

Reviewer # 1

1. This paper describes and approach to improving the flame retardancy of polyurethane foams by incorporating compounds that accelerate the decomposition of the urethane bond as well as oxidation of polyether polyols to generate carbon dioxide that extinguishes fire. Overall, the approach is very creative and doesn't rely on halogenated or phosphorous compounds. The authors have done significant experimental work to verify the reaction mechanisms and the conclusions are supported by the experimental data.

Author's response: Thank you for acknowledging the innovation and merits of our work. Your recognition has bolstered our confidence, inspiring us to further enhance and develop highly efficient flame-retardant systems.

2. A few issues to consider prior to publication: The "fire extinguisher" analogy seems weak and not especially scientific. A CO₂ fire extinguisher has pressurized CO₂ that is released. This system chemically generates CO₂. So other than CO₂ being emitted the analogy isn't very strong. I'd suggest removing it from the title and Figure 1.

Author's response: Thank you for your constructive suggestion. As you mentioned, a CO₂ fire extinguisher releases pressurized CO₂, while in this work, K-salts promote polyurethane foam to transform into much CO₂, increasing the CO₂ concentration in the gaseous products. According to your suggestion, the title has been revised as "Catalytic Polymer Self-Cleavage for CO₂ Generation Before Combustion Empowers Materials with Intelligent Fire Safety". Additionally, the updated Figure 1 now excludes the description of CO₂ fire extinguisher, providing a clearer representation of the K-salts catalytic material self-decomposition strategy for CO₂ production.

3. There are a number of phrases in English that are a bit awkward. (1) In the abstract (lines 32-33): "The incorporation of just 1.68 parts per hundred polyether polyols by weight..." This took me a while to figure out. Suggest: "The incorporation of just 1.68 parts per hundred by weight relative to polyether polyol of potassium formate..." (2) Line 55: should be "Lewis acids" (since retardants is plural, the acids has to be

plural) (3) Lines 62-63: Suggest: “The excessive addition of these retardants may compromise the overall material properties, particularly their mechanical properties.”

Author’s response: Thank you for kindly pointing out these imprecise expressions. According to your comment, these expressions have been corrected and marked blue in the revised manuscript. Additionally, the language of the manuscript has been carefully checked.

4. Line 91: “achieve high fire safe” Not sure what this means.

Author’s response: We are sorry for the spelling mistake. “achieve high fire safe” should be “achieve high fire safety”. In order to comprehensively demonstrate the elevated fire safety of K-salt filled FPUF, the revised manuscript incorporates the horizontal burning test, smoke toxicity test, and smoke density test. These evaluations aim to assess the flame spread rate, smoke toxicity, and smoke density. The results of these tests reveal that the inclusion of K-salts not only leads to a substantial improvement in limiting oxygen index but also effectively retards the flame spread rate. Furthermore, K-salts demonstrate a remarkable reduction in both smoke toxicity and density, highlighting their significant contribution to enhancing the overall fire safety characteristics of the material.

5. A major issue that should be remedied is that I was very confused at the outset as to what was actually being done. Figure 1a is very far from the section in the introduction that gives an overview. And then the results section just jumps in discussing the experiments. I think the authors should have a standalone figure (not part of figure 1) that gives a concrete overall perspective on what is being done to give the reader the proper context so they can understand what experiments are being carried out.

Author’s response: Thank you very much for your constructive suggestion. According to your comment, a redesigned standalone Figure 1 has been created to more effectively illustrate the strategy employed in this study. Furthermore, in order to enhance reader comprehension, the revised manuscript now includes explanations in the introduction section. These explanations cover the fire hazards associated with pure FPUF, along with details that we did about the design of K-salts catalyzing FPUF self-cleavage to proactively release CO₂ and reduce the concentration of toxic gases.

Figure 1. Schematic illustration of the catalytic polyurethane self-cleavage for CO₂ generation empowering the materials with fire safety. a) Diagram illustration of K-salt catalyzed polyurethane auto-pyrolysis to release CO₂ early for fire extinguishing. b) Pyrolysis processes of polyurethane catalyzed by K-salt.

6. Line 104: “The K-salts are introduced into the FPUF matrix during the in-situ foaming process.” Not sure I understand how this is done. I would think the additives need to be mixed into the formulation prior to the foaming process.

Author’s response: Thank you for kindly pointing out this imprecise expression. “The K-salts are introduced into the FPUF matrix during the in-situ foaming process” should be “The K-salts are pre-mixed with polyether polyols prior to the foaming process of FPUF”. The preparation process is shown in the following Supplementary Figure 1.

Supplementary Figure 1. Schematic diagram of the fabrication process for K-salt filled FPUF.

7. Line 123: Please define TTI.

Author’s response: Thank you for your valuable comment. TTI, defined as Time to Ignition, is a crucial parameter obtained from cone calorimetry to assess the fire safety of materials. According to your comment, the definition of TTI has been added to the revised manuscript.

Reviewer # 2

1. The authors present an interesting way to improve fire safety of PU foams by simple incorporation of K salts.

Author's response: Thank you for acknowledging the innovation and merits of our work. Your recognition has bolstered our confidence, inspiring us to further enhance and develop highly efficient flame-retardant systems.

2. As LOI is not a standard test for PU foam applications, I suggest to test the foams using UL94 HB test which is most common for automotive applications. Furthermore, LOI tests are very misleading as melting away of the foam from the applied flame can give wrong LOI values.

Author's response: Thank you for your thoughtful consideration and feedback. In response to your suggestion, we conducted the horizontal burning test (UL 94-HB) to assess the flame spread rate of PU foams. The results revealed a significantly high flame spread rate (63.0 ± 7.6 mm/min) for pure FPUF. While commonly used commercial expandable graphite (EG) and tris(1-chloro-2-propyl) phosphate (TCPP) slowed down the flame spread rate, they failed to achieve the highest rating (HF-1) due to issues such as a long-burned length or melting droppings igniting the bottom cotton. In contrast, **both K-formate and K-malate demonstrated the ability to self-extinguish FPUF, achieving the highest rating without any melting droppings** (refer to Supplementary Table 2 and Supplementary Video 4-8). These results underscore the exceptional flame retardancy endowed by K-salts in FPUF.

On the other hand, the specific advantage of the LOI test is that it provides numerical results and typically demonstrates linear relationships with the flame resistance level. As talking about, LOI tests can be misleading in some cases, as melting droppings may carry away heat from the burning surface, leading to inaccurate LOI values. In our study, both K-formate and K-malate filled FPUF showcased a notable absence of melting droppings during ignition. Furthermore, these materials could be promptly extinguished after removing the igniter, highlighting impressive LOI values of 26.5% and 26.0%, respectively.

Supplementary Table 2. Results of the horizontal burning test (UL 94-HB).

Samples	Burning velocity (mm/min)	Ignite absorbent cotton (Yes/No)	Rating
Pure FPUF	63.0 ± 7.6	Yes	NO
FPUF/EG	20.7 ± 3.7	No	HBF
FPUF/TCPP	/	Yes	HF-2
FPUF/K-formate	/	No	HF-1
FPUF/K-malate	/	No	HF-1

3. I am not sure if the authors can really claim that the fire performance is better for the modified foams, just from LOI values. In fact, the cone experiments provided in the supplementary info paint a different picture. The modified foams have higher HRR which indicates a higher risk in case of fire.

Author's response: We sincerely appreciate your thorough review and feedback. Following your suggestions, we have conducted more in-depth research, including the horizontal burning test, smoke toxicity, and smoke density tests, to provide a comprehensive understanding of the improved fire safety of the modified foams.

In the cone calorimetry analysis, some crucial parameters for evaluating material fire resistance, such as time to ignition (TTI), heat release rate, fire performance index (FPI), and fire growth index (FGI), were considered. Supplementary Table 5 demonstrates that the modified foams exhibit an over 800% increase in TTI compared to the pure sample, indicating enhanced resistance to ignition. This extension in ignition time is invaluable for creating additional moments for escaping and saving lives during fire incidents. While the modified foams may have a slightly higher peak of heat release rate (pHRR), their time to ignition and time to peak of heat release rate (t_p) are significantly prolonged. Consequently, the fire performance index ($FPI = TTI/pHRR$) of the modified foams increased by over 600%. A higher FPI implies a longer flashover time, providing people with more extended escape time. Additionally, the fire growth index ($FGI = pHRR/t_p$) of the modified foam was reduced by over 40%, effectively diminishing the risk of early-stage fire propagation. This underscores the merit of the flame-retardant system implemented in this study.

Supplementary Table 5. Characteristic data of different FPUF samples in the cone calorimetry.

Samples	TTI (s)	pHRR (kW/m ²)	t _p (s)	FPI (m ² s/kW)	FGI (kW/m ² /s)
Pure FPUF	11 ± 2	389 ± 37	62 ± 3	0.03 ± 0.01	6.27 ± 0.60
FPUF/K-formate	102 ± 3	488 ± 10	138 ± 3	0.21 ± 0.01	3.54 ± 0.07
FPUF/K-malate	105 ± 1	481 ± 1	130 ± 5	0.22 ± 0.01	3.70 ± 0.01

4. Please provide information such as smoke density and smoke toxicity (considering ammonia, HCN, and carbon monoxide evolution) for the PU foams.

Author's response: Thank you for your valuable feedback. Following your suggestions, we conducted smoke toxicity and smoke density tests to enhance the evaluation of FPUF fire safety. In fire incidents, inhalation of toxic smoke and gases contributes to 50-80% of fatalities. Polyurethane combustion produces harmful gases like carbon monoxide (CO), oxides of nitrogen (NO_x), and hydrogen cyanide (HCN). Unfortunately, traditional flame retardants, including tris(1-chloro-2-propyl) phosphate (TCPP), tend to increase polyurethane smoke toxicity. Figure 2g illustrates that TCPP addition notably elevates CO and HCN production.

Conversely, both K-formate and K-malate substantially reduce CO, HCN, and NO_x concentrations. Particularly, K-salt filled FPUFs exhibit over 60% reduction in HCN production. K-formate leads to a remarkable decrease in NO_x concentrations from 96.3 ppm to 1.8 ppm, well below the immediately dangerous to life and healthy (IDLH) concentration of NO_x (20 ppm) per the National Institute of Occupational Safety and Health (NOISH). Consequently, the conventional toxicity index (CIT_G) of FPUFs filled with K-formate and K-malate decreases by 95% and 86%, respectively, demonstrating the smoke toxicity inhibitory effect of K-salts (Figure 2h and Supplementary Table 7). Additionally, as indicated in Supplementary Table 8, the incorporation of K-salts leads to a substantial reduction in smoke density (approximately 20) in FPUF. This is notably lower compared to pure foam (83) or foam with TCPP (107). With a high LOI value, low flame spread rate, and diminished smoke toxicity and density, K-salts filled FPUF exhibits outstanding fire safety.

Figure 2. g) Concentrations of CO, HCN and NO_x in smoke toxicity tests. h) General conventional index of toxicity obtained from smoke toxicity tests.

Supplementary Table 7. Smoke toxicity gas concentration and general conventional index of toxicity (CIT_G) obtained from smoke toxicity tests for the different samples.

Samples	Pure FPUF	FPUF/TCPP	FPUF/K-formate	FPUF/K-malate
CO/(ppm)	67.8 ± 1.2	75.6 ± 9.5	67.7 ± 4.1	55.1 ± 6.6
HCN/(ppm)	5.1 ± 0.1	5.5 ± 0.3	1.9 ± 0.7	1.6 ± 0.4
NO _x (ppm)	96.3 ± 2.8	51.3 ± 3.3	1.8 ± 0.9	12.1 ± 1.8
CIT _G	0.381 ± 0.007	0.219 ± 0.015	0.019 ± 0.003	0.053 ± 0.031

Supplementary Table 8. Results of the smoke density (Ds) test.

Samples	Ds, 1.5
Pure FPUF	83.2 ± 4.7
FPUF/TCPP	107.8 ± 6.6
FPUF/K-formate	23.3 ± 1.5
FPUF/K-malate	19.8 ± 1.5

Reviewer # 3

1. Authors investigated the role of potassium salts as flame retardants for polyurethane foams acting through catalytic polymer autopyrolysis mode. Although it is claimed that “Here, we introduce an innovative strategy for the first time...” the autocatalytic reactive flame retardants (ACRFR) concept has been applied for polyurethanes, e.g. Materials Chemistry and Physics, Volume 267, 15 July 2021, 124636; Polymer Degradation and Stability, Volume 177, July 2020, 109168, and references cited therein. Potassium salts, such as potassium carbonate derivatives have also been tested as PU foams flame retardants, e.g. INVESTIGATION OF EFFECTS OF ALUM AND POTASSIUM SESQUICARBONATE ON THE FIRE CHARACTERISTICS OF FLEXIBLE POLYURETHANE FOAM (<https://www.projectreserve.com/2020/04/investigation-of-effects-of-alum-and-potassium-sesquicarbonate-on-the-fire-characteristics-of-flexible-polyurethane-foam.html>). This work is methodologically sound, however the degree of novelty does not warrant publication in Nature Communication.

Author's response: Thank you for dedicating time to review our manuscript. We sincerely apologize for any inappropriate descriptions that may have led to misunderstandings and hindered a comprehensive assessment of our work's merits. Following a thorough examination of the three referenced published works, we have carefully revised our manuscript, aiming to eliminate potential sources of misunderstanding and underscore the novelty of our study. We seek to clarify the fundamental distinctions between our work and the three cited publications. We hope these revisions address concerns, and we will be happy to revise further if necessary. Your understanding and feedback are highly valued.

(1) The negative comments primarily stem from the comment that "autocatalytic reactive flame retardants have been reported" (Materials Chemistry and Physics, 2021, 267, 124636). This is a misunderstanding with fundamental differences. **The "autocatalysis" referred to in this reported work pertains to the catalysis of the preparation reaction of polyurethane foam.** Specifically, the flame retardants (OPDOPO and IDEDOPO) containing tertiary amine groups act as catalysts, facilitating the polymeric reaction between isocyanate and polyol or water in the formation of polyurethane foam. The improvement in flame retardancy still relies on traditional phosphorus-containing structure (DOPO), resulting in a limiting oxygen index (LOI) of only 22.7% for prepared rigid polyurethane foam at high loadings (20% OPDOPO). Toxic release is likely to worsen, although this is not mentioned in the article.

In contrast, in our work, the term "autocatalysis" refers to the catalysis of the decomposition of polyurethane foam at high temperatures or during combustion, leading to the generation of a substantial amount of CO₂ for efficient flame retardation. At an extremely low additive level (1.68%), potassium salts (potassium formate and potassium malate) with strong nucleophilicity significantly improve the LOI of flexible polyurethane foam to 26%, tremendously reduce general conventional index of smoke toxicity by 95%, and extend the time to ignition during large-scale fires by an impressive 900%.

Clearly, this concept of "catalytic material cleavage for CO₂ generation towards fire safety" is entirely different from the "catalyst for foam preparation" mentioned in the comparative literature, with only a similarity in names. We have modified relevant descriptions to avoid this misunderstanding.

Figure 1. Structure of flame retardants (OPDOPO and IDEDOPO) used in the contrast work #1.

(2) Similarly, contrast work #2 (Polymer Degradation and Stability, 2020, 177, 109168) reported that potassium acetate (KAc) acted as the catalyst in the preparation of polyurethane foam, promoting the generation of isocyanurate rings under high NCO index during PUF foaming. While this contributed to a reduction in the heat release rate of phosphorus-containing rigid polyisocyanurate foam to some extent,

phosphorus-containing flame retardants (EA-BPPO) still dominate in enhancing the flame retardancy of polyurethane. Obviously, this study does not relate to the core focus of our work, which centers on catalyzing the decomposition of polyurethane foam during combustion, leading to the substantial generation of CO₂ for efficient flame retardation.

In our study, KAc serves as a contrasting example, highlighting the novelty of our approach. It is revealed that KAc lacks the catalytic capability for polymer self-cleavage, hindering CO₂ generation. The LOI of KAc filled FPUF stands at only 22.5%, indicating weak flame retardancy. In contrast, potassium salts with strong nucleophilicity, such as potassium formate and potassium malate, demonstrate the ability to catalyze polyurethane self-cleavage, resulting in a significant volume of CO₂. This approach, devoid of traditional phosphorus/halogen elements, yields a high limiting oxygen index value, a low flame spread rate, as well as low smoke toxicity and density.

Figure 2. Structure of flame retardant (EA-BPPO) used in the contrast work #2.

(3) In the contrast work #3 (<https://www.projectreserve.com/2020/04/investigation-of-effects-of-alum-and-potassium-sesquicarbonate-on-the-fire-characteristics-of-flexible-polyurethane-foam.html>), two potassium-based inorganic fillers (alum and potassium sesquicarbonate) are discussed, utilizing their high-temperature degradation to absorb heat and release a small amount of non-combustible gas for flame retardation (Figure 3). This study does not relate to the core of our work, which is the catalytic effect of specific potassium salts on polyurethane decomposition for efficient flame retardation.

In fact, we also use potassium carbonate as a contrasting example to illustrate the novelty of our work. In our study, we focus on studying the nucleophilicity of potassium salts in catalyzing the self-cleavage of polyurethane to generate a substantial volume of carbon dioxide that extinguishes fire. The addition of potassium salts is kept very low, at less than 3%, ensuring that the pyrolysis products themselves are not sufficient to prevent the flame propagation of polyurethane. Based on our test results,

potassium sulfate and potassium carbonate are not active in catalyzing CO₂ release from the decomposition of FPUF because their anionic ligands lack nucleophilicity, resulting in their flame retardancy not being prominent. Only potassium salts with strong nucleophilicity, such as potassium formate and potassium malate, can catalyze the self-cleavage of polyurethane to generate a substantial volume of CO₂. This leads to a high limiting oxygen index value, a low flame spread rate, as well as low smoke toxicity and density.

Figure 3. Decomposition reactions of the flame retardants (alum and potassium sesquicarbonate) used in the contrast work #3.

Consequently, the three published works do not diminish the innovation of our study but rather underscore its originality. We acknowledge that any misunderstanding may have arisen from unclear descriptions in our manuscript. In response, we have made revisions, including the removal of terms such as "autocatalytic" with the aim of preventing any potential confusion. We sincerely hope that these adjustments to our manuscript effectively highlight the innovation of our work and address any concerns. Your satisfaction with the revised version is our primary goal.

Reviewers' Comments:

Reviewer #2:

Remarks to the Author:

The authors have fully addressed my comments and it can now be accepted for publication.